# Efficient Pure Exploration in Adaptive Round model

**Tianyuan Jin**[†], **Jieming Shi**[‡], **Xiaokui Xiao**[‡], **Enhong Chen**[†*]
[†]School of Computer Science and Technology, University of Science and Technology of China
[‡]School of Computing, National University of Singapore
[†] jty123@mail.ustc.edu.cn, cheneh@ustc.edu.cn, [‡]{shijm, xkxiao}@nus.edu.sg

## Abstract

In the adaptive setting, many multi-armed bandit applications allow the learner to adaptively draw samples and adjust sampling strategy in rounds. In many real applications, not only the query complexity but also the round complexity need to be optimized. In this paper, we study both PAC and exact top-$k$ arm identification problems and design efficient algorithms considering both round complexity and query complexity. For PAC problem, we achieve optimal query complexity and use only $O(\log^*_{\frac{k}{\delta}}(n))$ rounds, which matches the lower bound of round complexity, while most of existing works need $\Theta(\log \frac{n}{k})$ rounds. For exact top-$k$ arm identification, we improve the round complexity factor from $\log n$ to $\log^*_{\frac{1}{\delta}}(n)$, and achieve near optimal query complexity. In experiments, our algorithms conduct far fewer rounds, and outperform state of the art by orders of magnitude with respect to query cost.

## 1 Introduction

Mutli-armed bandit (MAB) problems are classic decision problems with numerous applications such as medical trials [1], online advertisement [2], and crowdsourcing [3]. These problems typically consider a bandit with a set of arms, each of which has an unknown reward distribution with an unknown mean, and the objective is either to (i) identify the top-$k$ arms with the maximum reward means or (ii) maximize the expected total reward under some constrains on the costs of arm pulling.

This paper studies the problem of top-$k$ arms identification in the *adaptive* setting, which allows the leaner to draw samples from the arms adaptively in *rounds* to estimate their means, and to adjust the sampling strategy for the $i$-th round based on the observations from the first $i - 1$ rounds. Following previous work [4], we assume that in each round, the learner is allowed to query an arbitrary number of arms for an arbitrary number of times, but the query results would only be revealed at the end of the round. We aim to minimize the number of rounds performed, as well as achieving best possible query complexity. In addition, our proposed algorithms exhibit superior practical performance due to our small constant factors. Existing top-$k$ algorithms mainly focus on query complexity and most of them are not efficient due to their large constants [5, 6, 3, 7, 8] or inferior query complexities [9, 10]. Adaptive round setting of MAB has many real applications, as described below.

**Medical trials.** In medical trials [11], to identify the best drug for a disease, one can conduct tests in rounds, such that each round involves testing multiple candidate drugs on multiple clinical subjects (e.g., mice) simultaneously. However, after each round of testing, there is typically a waiting time (e.g., days) before the effects of drugs become observable to guide the design of the next round of testing. It is important to minimize not only the total number of tests on clinical subjects (i.e., query complexity) but also the number of rounds, to identify the best drug within the shortest time frame.

---

[*]Corresponding author

**Online advertisement.** In online advertisement [12], an advertiser may push ads to the users of candidate websites, so as to identify the top-$k$ websites that have the highest click-through rates and match some clients advertising requirements. The pushing of ads could be conducted in rounds, and each round may involve multiple ads and multiple users. However, in each round, it takes time to observe users' responses to the ads, and to decide which websites are unpromising and should be pruned in the next round. In this application, there is usually a tight time frame to offer a solution to the clients, so as to ensure the timeliness of the ads.

**Crowdsourcing.** Workers on crowdsourcing platforms often vary significantly in terms of the answer quality. As an effective strategy to identify the most reliable workers for a specific task, one may test each worker with a sequence of questions with ground-truths, and then select workers based on the accuracy of their answers. Note that for such tests, workers need some time to answer the questions, and need to be rewarded upon the completion of the questions. To minimize the time and monetary cots, it is crucial to have an algorithm to identify the most reliable workers that minimizes the number of tests (i.e., query complexities) within a limited number of rounds, where our proposals fit.

## 1.1 Problem Formulation

Under the standard setting of stochastic multi-armed bandit selection, there is a set $S$ of $n$ arms, such that each arm $i$ is associated with an unknown reward distribution $\mathcal{D}_i$ supported on $[0, 1]$ with unknown mean $\theta_i$. Let $i^*$ be the arm with $i^{th}$ largest mean. We aim to identify the $k$ arms with the largest means by pulling (i.e., sampling from) the arms in rounds. In each round, we can pull any number of arms for any number of times, such that (i) each pull of an arm $i$ returns a reward that is an i.i.d. sample from $\mathcal{D}_i$, and (ii) the reward is only revealed at the end of the round.

For *PAC subset selection*, we study two problems: (i) Problem 1 (PAC-top-$k$): PAC Top-$k$ Arm Selection with Adaptive Rounds, and (ii) Problem 2 (RL-top-$k$): Top-$k$ Arm with a Round Limit $R$. In both problems, the goal is to identify a set $V \subseteq S$ of $k$ arms, such that for all $i \in [1, k]$, the $i^{th}$ largest arm in $V$ has mean larger than $\theta_{i^*} - \epsilon$ with probability at least $1 - \delta$, where $\epsilon$ and $\delta$ are given constants. Specifically, for PAC-top-$k$, we aim to minimize the number of rounds performed, while achieving the best possible query complexity; for RL-top-$k$, we expose a upper limit on the number of rounds that can perform, $R$, and aim to minimize the query complexity within $R$ rounds.

For *exact top-$k$ arm identification*, denoted as Problem 3 (exact-top-$k$), we aim to minimize the number of rounds required as well as the query cost, for identifying the top-$k$ arms with the largest means. We assume $\theta_{k^*} > \theta_{(k+1)^*}$, in order to ensure the uniqueness of the solution.

## 1.2 State of the Art

To the best of our knowledge, Agarwal et al.'s work [4] is the only one that studies the top-$k$ arms problem while taking into account the round complexity. In particular, [4] studies the identification of exact top-$k$ arms with adaptive rounds, and presents a method that takes $\Delta_k$ as input and returns the exact top-$k$ arms with at least $1 - \delta$ probability, with query complexity $O\left(\frac{n}{\Delta_k^2} \cdot \log \frac{k}{\delta}\right)$ and round complexity[2] $\log^*(n)$ , where $\Delta_k$ denotes the difference between the means of the $k^{th}$ and $(k+1)^{th}$ largest arms, and $\log^*(n)$ denotes the *iterated logarithm* of $n$, i.e.,

$$\log^*(n) = \begin{cases} 1 + \log^*(\log n), & \text{if } n > 1 \\ 0, & \text{otherwise} \end{cases} \tag{1}$$

In other words, $\log^*(n)$ equals the number of times that we need to apply the logarithm function on $n$ before the result is no more than 1. Furthermore, [4] also studies the problem where the round limit $R$ is given. Their algorithm identifies the exact top-$k$ arms with at least $1 - \delta$ probability, with a query complexity of $O\left(\frac{n}{\Delta_k^2} \cdot (\log \frac{k}{\delta} + \mathrm{ilog}^{(R)}(n))\right)$, where $\mathrm{ilog}^{(R)}(n)$ is the result of iteratively applying the logarithm function on $n$ for $R$ times, i.e.,

$$\mathrm{ilog}^{(r)}(x) = \begin{cases} \mathrm{ilog}^{(r-1)}(\log(x)), & \text{if } x > 1 \\ 1, & \text{if } x \le 1. \end{cases} \tag{2}$$

With respect to lower bound, Agarwal et al. show that a round complexity of $\log^*(n)$ is near optimal, since for constants $k$ and $\delta$, any algorithm with $O(\frac{n}{\Delta_k^2})$ query complexity requires at least

| | Algorithm | Number of Rounds | Query Complexity |
|---|---|---|---|
| $k=1$ | [5] | $\Theta(\log n)$ | $O\left(\frac{n}{\epsilon^2}\cdot\log\frac{1}{\delta}\right)$ |
| All $k\in[n]$ | [6, 16, 14] | $\Theta(\log\frac{n}{k})$ | $O\left(\frac{n}{\epsilon^2}\cdot\log\frac{k}{\delta}\right)$ |
| | This paper (Algorithm 1) | $2\log^*_{\frac{k}{\delta}}(n)$ | $O\left(\frac{n}{\epsilon^2}\cdot\log\frac{k}{\delta}\right)$ |

Table 1: Summary of algorithms for Problem 1: Top-$k$ arms with adaptive rounds.

| | Algorithm | Bound | Query Complexity |
|---|---|---|---|
| All $k\in[n]$ | [4], assuming $\Delta_k$ is known | exact top-$k$ | $O\left(\frac{n}{\Delta_k^2}\cdot\left(\log\frac{k}{\delta}+\mathrm{ilog}^{(R)}(n)\right)\right)$ |
| | This paper (Algorithm 2) | $(\epsilon,\delta)$ | $O\left(\frac{n}{\epsilon^2}\cdot\left(\log\frac{k}{\delta}+\mathrm{ilog}^{(R)}_{\frac{k}{\delta}}(n)\right)\right)$ |

Table 2: Summary of algorithms for Problem 2: Top-$k$ arms with a round limit $R$

.

$\log^*(n)-\log^*(\Theta(\log^*(n)))$ rounds. Besides, Agarwal et al. prove that identifying the exact top-$k$ arms with at least $3/4$ probability using $R$ rounds must use $\Omega\left(\frac{n}{\Delta_k^2 R^4}\cdot\mathrm{ilog}^{(r)}\left(\frac{n}{k}\right)\right)$ samples.

Agarwal et al.'s algorithm suffers from a major deficiency that it requires $\Delta_k$ to be known in advance, which is unrealistic in most practical applications as the mean of each arm is unknown. In addition, the algorithm cannot be extended to address PAC-top-$k$ and RL-top-$k$ by replacing $\Delta_k$ with $\epsilon$, since the algorithm strongly relies on the assumption that there is exact $k$ arms whose means are larger than $\theta_{k^*}-\Delta_k$, where $k^*$ is the arm with the $k^{th}$ largest mean. (This assumption does not hold in general if we replace $\Delta_k$ with any $\epsilon>\Delta_k$.) Further, the algorithm cannot be used to get instance-dependent query complexity (where the query complexity not only depends on $\Delta_k$ but also depends on $\{\theta_i\}_{i=1}^n$), since all Exponential-Gap-Elimination algorithms [8, 13, 14, 15] need a PAC algorithm as a subroutine.

There also exists a number of techniques [16, 6, 13, 10, 14, 8] for both PAC and exact top-$k$ arm identification problems that optimizes the query complexity, without considering the round complexity. The query complexity achieved by these technique is near optimal. However, all of these incur $\log n$ factor on round complexity, significantly worse than the round complexity of [4].

### 1.3 Our Results

In this paper, we present three algorithms for the top-$k$ arm selection problems in adaptive round model. Below summarizes our results.

**Theorem 1.** *There is an algorithm that computes $\epsilon$-top-$k$ arms with probability at least $1-\delta$, pulls the arms at most $O(\frac{n}{\epsilon^2}\cdot\log\frac{k}{\delta})$ times and runs in at most $2\log^*_{\frac{k}{\delta}}(n)$ expected rounds.*

**Theorem 2.** *There is an algorithm that computes $\epsilon$-top-$k$ arms with probability at least $1-\delta$, pulls the arms at most $O(\frac{n}{\epsilon^2}\cdot(\mathrm{ilog}^{(R)}_{\frac{k}{\delta}}(n)+\log\frac{k}{\delta}))$ times and runs within $R$ rounds.*

Since (i) the solution in [4] is proved to be near-optimal, and (ii) the problems studied in [4] are special cases of PAC-top-$k$ and RL-top-$k$ with $\epsilon\leftarrow\Delta_k$, the round complexity of our algorithm for PAC-top-$k$ and the query complexity of our algorithm for $RL$-top-$k$ are near-optimal.

Compared with the solution in [4], our algorithms do not require any prior knowledge of $\Delta_k$, and allow us to choose an error parameter $\epsilon\in(0,1)$ to strike a trade-off between the accuracy and efficiency of the algorithm, which is much more practical. Further, our PAC version can be used to get instance-dependent query complexity while [4] can not.

**Theorem 3.** *There is an algorithm that computes exact top-$k$ arms with probability at least $1-\delta$, pulls the arms at most $O\left(\sum_{i=1}^n\Delta_i^{-2}\log\left(\frac{k\cdot\log\Delta_i^{-1}}{\delta}\right)\right)$ times and runs in $O(\log^*_{\frac{1}{\delta}}n\cdot\log\Delta_k^{-1})$ rounds.*

Compared with the previous exact top-$k$ arm algorithms [14, 8, 13], we improve the factor on round complexity from $\log n$ to $\log^*_{\frac{1}{\delta}}(n)$, while achieving the same query complexity. Tables 1, 2 and 3 summarize our results and those of the state-of-the-art methods.

## 2 PAC Subset Selection

We present our algorithms for the PAC top-$k$ arms selection problems, i.e., PAC-top-$k$ and RL-top-$k$.

| | Algorithm | Round Complexity | Query Complexity |
|---|---|---|---|
| $k = 1$ | [8] | $O(\log n \cdot \log \Delta_k^{-1})$ | $O\left(\sum_{i=1}^n \Delta_i^{-2} \cdot \log \frac{\log \Delta_i^{-1}}{\delta}\right)$ |
| | [17] | $O\left(\sum_{i=1}^n \Delta_i^{-2} \cdot \log \frac{\log \Delta_i^{-1}}{\delta}\right)$ | $O\left(\sum_{i=1}^n \Delta_i^{-2} \cdot \log \frac{\log \Delta_i^{-1}}{\delta}\right)$ |
| All $k \in [n]$ | [10] | $O\left(\sum_{i=1}^n \Delta_i^{-2} \cdot \log \frac{\sum_{i=1}^n \Delta_i^{-1}}{\delta}\right)$ | $O\left(\sum_{i=1}^n \Delta_i^{-2} \cdot \log \frac{\sum_{i=1}^n \Delta_i^{-1}}{\delta}\right)$ |
| | [13] | $O(\log n \cdot \log \Delta_k^{-1})$ | $O\left(\sum_{i=1}^n \Delta_i^{-2} \cdot \log \frac{k \cdot \log \Delta_i^{-1}}{\delta}\right)$ |
| | This paper | $O(\log^*_{\frac{1}{\delta}} n \cdot \log \Delta_k^{-1})$ | $O\left(\sum_{i=1}^n \Delta_i^{-2} \cdot \log \frac{k \cdot \log \Delta_i^{-1}}{\delta}\right)$ |

Table 3: Summary of algorithms for Problem 3: Exact top-$k$ arm identification. (For $i \le k$, $\Delta_i$ denotes the difference between the means of the $i^{th}$ and $(k+1)^{th}$ arms. For $i > k$, $\Delta_i$ denotes the difference between the means of the $k^{th}$ and $i^{th}$ arms.)

## 2.1 Top-$k$ $\delta$-Elimination

$k$-$\delta$E (Algorithm 1) can identify the top-$k$ arms for PAC-top-$k$, with query complexity $O(\frac{n}{\epsilon^2} \log \frac{k}{\delta})$ and at most $2 \log^*_{\frac{k}{\delta}}(n)$ expected rounds. Compared with Median Elimination based top-$k$ algorithms, e.g., [16, 6], which only eliminate half of the candidates in each round, $k$-$\delta$E can eliminate at least $100(1 - \frac{\delta}{k})$ percent of candidate arms every other round, which is far better. We go through the algorithm first and then explain why. Note that in each while iteration (Line 5-17), $k$-$\delta$E performs two separate rounds of pulling (Line 5 and Line 8), since the pulls at Line 8 are dependent on the empirical results obtained at Line 5. This corresponds to the 2 factor in our round complexity $2 \log^*_{\frac{k}{\delta}}(n)$. Without ambiguity, $r$ means iterations in Algorithm 1, but means rounds in Algorithm 2.

Algorithm 1 takes as input $S, Q, k, \epsilon, \delta$, where $S$ is the set of all the arms and $Q = \frac{c}{\epsilon^2}$ ($c$ is an constant factor determined in Lemma 1). An empty set $S'$ (Line 3) is initialized for the storage of the arms and their empirical means obtained later in the algorithm. In each iteration, we pull every arm in $S_r$ by $Q_r$ times, and sort them by their empirical means (Line 5). At Line 7-8, we *double test* the empirical mean of each arm in $S_{k'}^r$ (in order to keep the estimation unbiased) and keep it in $S'$. Then we update $S_r$ to $S_{r+1}$ by only keeping the arms with empirical means $3/4\epsilon$ greater than the $k^{th}$ largest mean in $S'$, and also excluding the arms in $S_{k'}^r$ (Line 10). From Line 11 to 15, we update $\beta_r$ and $\delta_r$, which can make $Q_r$ exponentially decrease in next iteration. This is critical to keep the total number of pulls linear to $n$. The whole process continues until $S_r$ is empty, then the top-$k$ arms in $S'$ are returned.

Median Elimination (ME) methods can only allow $\epsilon_r$ regret in each iteration ($\sum_r \epsilon_r \le \epsilon$), in order to guarantee $\epsilon$ error bound even when the best arm is mistakenly eliminated. On the other hand, $k$-$\delta$E allows $\epsilon$ loss in each iteration with the help of $S'$ and *double test*, which allows us to perform fewer pulls and eliminate more than half of the arms per iteration. During a iteration $r$, ME methods need to sample $O((1/\epsilon_r^2) \log(k/\delta_r))$ times per arm, much larger compared to $O((1/\epsilon^2) \log(k/\delta_r))$. It is even worse when $r$ increases (i.e., $\epsilon_r$ decreases), leading to the large constant factors in ME methods.

Specifically, in Algorithm 1, $S'$ stores randomly chosen arms that are eliminated. It holds that the top $i^{th}(i \le k)$ arm stored in $S'$ is at most $\epsilon$ smaller than $i^{th}$ eliminated arm (see the proof details of Lemma 1). If $k$-$\delta$E has eliminated the $i^{th}$ largest arm, then with high probability the $i^{th}$ largest arm stored in $S'$ must be the $\epsilon$-approximate of $i^{th}$ largest arm. Hence, $k$-$\delta$E allows $\epsilon$ loss per iteration.

Moreover, $k$-$\delta$E uses a more aggressive indicator to eliminate arms, compared to the median indicator used in Median Elimination based algorithms. We use as our indicator, the $k^{th}$ largest empirical mean of the randomly chosen top arms stored in $S'$, plus $3/4\epsilon$ (Line 10 of Algorithm 1). However, directly using such indicator without double test, the indicator may be positively biased. And then all the $\epsilon$-top-$k$ arms might be eliminated with such indicator, which leads to wrong results. To deal with this, we use double-test strategy to re-sample another $Q_r$ times at Line 8 before using the indicator at Line 10 in Algorithm 1, to keep the indicator unbiased. Further, $3/4\epsilon$ increment is added to the indicator to eliminate more arms safely, proved in Lemma 1. $k$-$\delta$E runs in $2 \log^*_{\frac{k}{\delta}}(n)$ expected rounds.

Compared to [4], our Algorithm 1 and Algorithm 2 are fundamentally different. We assume no prior knowledge of the arms, e.g., $\Delta_k$. Given $\Delta_k$, Agarwal et al.'s algorithm can compute an optimal indicator to eliminate the arms definitely not in top-$k$. Our indicator (Line 10) is set with the help of $S'$ and double test, which gives our algorithm near-optimal round complexity $2 \log^*_{\frac{k}{\delta}}(n)$.

**Algorithm 1** Top-$k$ $\delta$-Elimination ($k$-$\delta$E)

1: **Input:** $S$, $Q$, $k$, $\varepsilon$ and $\delta$.
2: Initialize $r \leftarrow 1$, $\beta_1 \leftarrow 1$, $\delta_1 \leftarrow \delta/4$, $S_1 \leftarrow S$.
3: Initialize $S' \leftarrow \varnothing$.
4: **while** $S_r \neq \varnothing$ **do**
5:      Sample each arm $i \in S_r$ for $Q_r \leftarrow \beta_r \cdot Q \cdot \log(\frac{k}{\delta_r})$ times; sort them decreasingly by empirical means $\hat{\theta}_i$;
6:      $k' \leftarrow \min\{k, |S_r|\}$;
7:      Uniformly sample $k'$ arms from the top-$[\lceil (\delta_r/k)^{\beta_r} \cdot |S_r|/2 \rceil + k' - 1]$ sorted arms as set $S_{k'}^r$;
8:      For each arm $i \in S_{k'}^r$, double test by re-sampling it $Q_r$ times and insert its new empirical mean into $S'$;
9:      Get the $k$-th largest mean in $S'$ as $S'(k)$;
10:      Set $S_{r+1} \leftarrow \{i \in S_r : \hat{\theta}_i \geq S'(k) + 3\epsilon/4\}$ and $S_{r+1} \leftarrow S_{r+1} \backslash S_{k'}^r$;
11:      **if** $|S_{r+1}| \leq \frac{2\delta}{k}|S_r|$ **then**
12:         $\beta_{r+1} \leftarrow \beta_r \frac{|S_r|}{2|S_{r+1}|}$;
13:      **else**
14:         $\beta_{r+1} \leftarrow \beta_r \frac{|S_r|}{|S_{r+1}|}$;
15:      **end if**
16:      $\delta_{r+1} \leftarrow \delta/(2 \cdot 2^r)$;
17:      $r \leftarrow r + 1$;
18: **end while**
19: **Return:** Top-$k$ arms in $S'$.

## 2.2 Bounding the Regret, Query Complexity, and Round Number of $k$-$\delta$E

We bound the regret in $k$-$\delta$E and give its query and round complexity. The proofs are in Appendix B.

**Lemma 1.** *Given a $n$-arm set, $S$, parameter $\epsilon \in (0,1)$, and $\delta \in (0, 1/4)$, it suffices to run Algorithm 1 with $Q \geq \frac{32}{\epsilon^2}$ in order to obtain a $k$-sized subset $V \subseteq S$, such that with probability at least $1 - \delta$, the $i^{th}$ largest arm in $V$ has mean larger than $\theta_{i*} - \epsilon$, for all $i \in [1, k]$. Additionally, if we change the parameter $3/4\epsilon$ (Line 10 in Algorithm 1) to $\epsilon_1$, where $\epsilon_1 \in (0, \epsilon)$, then by setting $Q \geq \frac{2}{(\epsilon - \epsilon_1)^2}$, Algorithm 1 still works with the $(\epsilon, \delta)$ guarantee.*

Lemma 1 provides the $(\epsilon, \delta)$ guarantee of algorithm $k$-$\delta$E. Lemma 2 shows that, *w.h.p.*, $S_{r+1}$ is $(\delta/k)^{-\beta_r}$ times smaller than $S_r$, which is used in Lemma 3 to bound the round complexity.

**Lemma 2.** *If $Q \geq \frac{57}{\epsilon^2}$ and $\delta \in (0, 1/4)$, then at iteration $r$, with probability at least $1 - 2\delta_r$, $|S_{r+1}| \leq \lceil 2 \cdot (\delta_r/k)^{\beta_r} |S_r| \rceil - 1$.*

**Lemma 3.** *For $Q \geq 57/\epsilon^2$ and $\delta \in (0, 1/4)$, with probability at least $1 - \delta$, the number of rounds $R'$ used in $k$-$\delta$E satisfies: $R' \leq 2 \log^*_{\frac{k}{\delta}}(n)$, and $\mathbf{E}[R'] \leq 2(1 + 2\delta) \log^*_{\frac{k}{\delta}}(n)$.*

Next, we provide the query complexity of $k$-$\delta$E in Lemma 4. Kalyanakrishnan et al. [10, Theorem 8] present a lower bound of $\Omega(\frac{n}{\epsilon^2} \log \frac{k}{\delta})$ for PAC version (the Explore-$k$ metric, see Section 5). Hence, up to a small constant factor, our query complexity is optimal. Combining Lemma 1,3, and 4, Theorem 1 follows.

**Lemma 4.** *Let $N$ be the number of arms pulled by Algorithm 1. For $Q \geq \frac{57}{\epsilon^2}$, with probability at least $1 - \delta$, $N \leq 7n \cdot Q \cdot \log(4k/\delta)$; and $\mathbf{E}[N] \leq 7(n + 1) \cdot Q \cdot \log(4k/\delta)$.*

**Remark 1.** In previous work [16], as the theoretical analysis is rather pessimistic due to the extensive usage of the union bound, the constant to achieve regret bound are far from tight. The constant can be even up to $10^5$(i.e., the bound is $N \geq \frac{10^5}{\epsilon^2} \log \frac{1}{\delta}$ in previous works).

In $k$-$\delta$E, (i) our constants (Lemma 4) are much smaller; (ii) our constant factor is adjustable according to Lemma 1 with the $\epsilon$ regret bound still guaranteed (for instance, the $3/4\epsilon$ factor in Line 10 of Algorithm 1 can be changed to $1/2\epsilon$, then setting $Q \geq \frac{8}{\epsilon^2}$ still guarantees the PAC bound); (iii) our algorithm can stop as soon as it is confident to find the correct arm, reducing the practical query cost.

## 2.3 Top-$k$ Arm Selection with a Round Limit

In this section, we propose $k$-$\delta$ER (Algorithm 2) to solve Problem 2, top-$k$ arm selection with a round limit $R$. Our proposal can report correct result within $R$ rounds, with almost optimal query complexity. Compared to $k$-$\delta$E, $k$-$\delta$ER only requires one round (Line 4) per iteration, rather than

---

**Algorithm 2** Top-$k$ $\delta$-Elimination with Limited Rounds ($k$-$\delta$ER)

---
1: **Input:** $S$, $R$, $k$,$Q$, $\epsilon$ and $\delta$.
2: Initialize $r \leftarrow 1$, $\delta_1 \leftarrow \delta/4$, $\beta_1 \leftarrow 1 + \mathrm{i}\log_{\frac{k}{\delta}}^{(R)}(n)$, $S_1 \leftarrow S$, $S' \leftarrow \varnothing$.
3: **for** $r \leq R - 1$ **do**
4:      Sample each arm in $S_r$ by $Q_r \leftarrow \beta_r \cdot Q \cdot \log(k/\delta_r)$ times, and sort decreasingly by their empirical $\hat{\theta}_i$;
5:      $k' \leftarrow \min\{k, |S_r|\}$;
6:      Uniformly sample $k'$ arms from the top-$[\lceil (\delta_r/k)^{\beta_r} \cdot |S_r|/2 \rceil + k' - 1]$ sorted arms as set $S_{k'}^r$;
7:      Add $S_{k'}^r$ into $S'$;
8:      Let $S_{r+1}$ be set containing all the top-$[\lceil 2 \cdot (\delta_r/k)^{\beta_r} |S_r| \rceil + k' - 1]$ sorted arms in $S_r$;
9:      $S_{r+1} \leftarrow S_{r+1} \backslash S_{k'}^r$;
10:     $\beta_{r+1} \leftarrow \beta_r \frac{|S_r|}{2|S_{r+1}|}$;
11:     $\delta_{r+1} \leftarrow \delta/(2 \cdot 2^r)$;
12:     $r \leftarrow r + 1$;
13: **end for**
14: **Return:** $\mathrm{US}(S', S_R, Q, \beta_R, \delta, k)$.

---

---

**Algorithm 3** Uniformly Sampling (US)

---
1: **Input:** $S', S_R, Q, \beta_R, \delta, k$.
2: Sample each arm $i \in S_R$ by $Q \cdot \beta_R \cdot \log \frac{2k \cdot 2^R}{\delta}$ times and sort decreasingly by their empirical $\hat{\theta}_i$;
3: Let $S_k^R$ be the set of all the top-$\min\{k, |S_R|\}$ arms;
4: Sample each arm $i \in S'$ by $Q \log \frac{4|S'|}{\delta}$ times, and let $\hat{\theta}_i$ be its empirical mean;
5: **Return:** Top-$k$ arms in $S_k^R \cup S'$.

---

two rounds per iteration. According to Lemma 2, in each iteration, we can bound the total number of arms in $S_{r+1}$ without double test. Thus, rather than performing the double test immediately (Line 8, Algorithm 1), $k$-$\delta$ER delays all the double-tests of all the iterations until the final round (Line 14, Algorithm 2), and conducts all the double-tests in this round, using uniform sampling (Algorithm 3).

Algorithm 2 shows the pseudo-code of $k$-$\delta$ER. It takes as input one more parameter, $R$, the round limit. $S'$ stores all the arms delayed for double test in the first $R - 1$ rounds (Line 7). At Line 14, Algorithm 3 is called to sample all the arms in both $S'$ and $S_R$ in one round, and then the the top-$k$ arms are reported. Note that in Algorithm 3, the samples in Line 2 and 4 can be submitted simultaneously, so this only cost one round. Compared to Median Elimination algorithms, $k$-$\delta$ER has similar advantages as $k$-$\delta$E analyzed in Section 3.1. With the help of $S'$ and double test, $k$-$\delta$ER can eliminate more arms in each round, while still provides $(\epsilon, \delta)$ guarantee, as follows.

**Lemma 5.** *Given a $n$-arm set, $S$, parameters $k$, $\epsilon \in (0, 1)$, $\delta \in (0, 1/4)$, and $1 \leq R \leq \log_{\frac{k}{\delta}}^{*}(n)$, it suffices to run Algorithm 2 with $Q \geq \frac{57}{\epsilon^2}$ in order to obtain a $k$-sized subset $V \subseteq S$, such that with probability at least $1 - \delta$, the $i^{th}$ largest arm in $V$ has mean larger than $\theta_{i^*} - \epsilon$, for all $i \in [1, k]$.*

Details of the proof are in Appendix B. Lemma 6 bounds the query complexity of $k$-$\delta$ER. When $R \geq \log_{\frac{k}{\delta}}^{*}(n)$, our algorithm can achieve the optimal query complexity using just $\log_{\frac{k}{\delta}}^{*}(n)$ rounds. Combining Lemma 5 and 6, Theorem 2 follows.

**Lemma 6.** *If $Q \geq \frac{57}{\epsilon^2}$, with target number of rounds $1 \leq R \leq \log_{\frac{k}{\delta}}^{*}(n)$, Algorithm 2 uses $O\left(\frac{n}{\epsilon^2}\left(\mathrm{i}\log_{\frac{k}{\delta}}^{(R)}(n) + \log(k/\delta)\right)\right)$ samples.*

## 3 Exact Top-$k$ Arm Identification

Here we solve exact-top-$k$ to identify exact top-$k$ arms. Our algorithm uses the Exponential-Gap-Elimination algorithm(e.g., [13, 14, 8]) as a framework, and uses Algorithm 1 as a component. Specifically, we replace the Median Elimination Algorithm used in [13, 14, 8] by Algorithm 1 and then prove the newly algorithms satisfies Theorem 3. Here, we use [13] as an example. In [13], it has three subroutines, called PAC-Best-$k$, EstMean-Large, EstMean-Small. We replace all of these subroutines by Algorithm 1, to get our algorithm for exact-top-$k$, denoted as Algorithm 4. In [13],

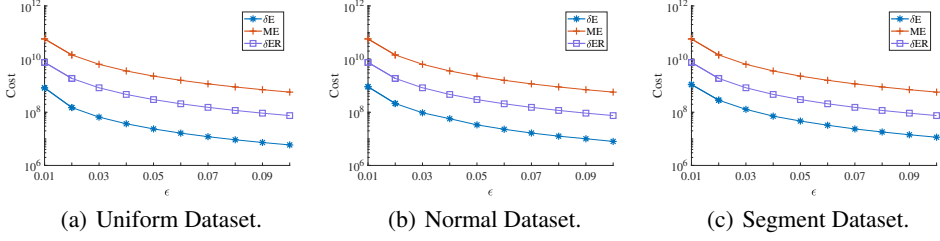

(a) Uniform Dataset.     (b) Normal Dataset.     (c) Segment Dataset.

Figure 1: Query cost of PAC best arm selection.

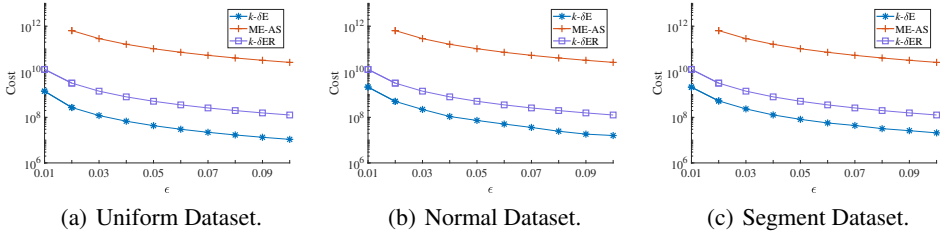

(a) Uniform Dataset.     (b) Normal Dataset.     (c) Segment Dataset.

Figure 2: Query cost of PAC top-$k$ arm selection.

it is proved that the query complexity is no worse than $O\left(\sum_{i=1}^{n}\Delta_i^{-2}\cdot\log\frac{k\cdot\log\Delta_i^{-1}}{\delta}\right)$. Combining Lemma 7 and 8, we get Theorem 3. The proofs are in Appendix B.

**Lemma 7** ([13], Theorem 1.2). *Algorithm 4 returns the correct answer with probability at least $1-\delta$ and takes $O\left(\sum_{i=1}^{n}\Delta_i^{-2}\cdot\log\frac{k\cdot\log\Delta_i^{-1}}{\delta}\right)$ samples.*

**Lemma 8.** *Algorithm 4 runs in $O\left(\log^*_{\frac{k}{\delta}}(n)\cdot\log\Delta_k^{-1}\right)$ rounds.*

## 4 Experiments

### 4.1 Experimental Results for PAC Top-$k$ Identification

For PAC top-$k$ arms, we compare $k$-$\delta$E and $k$-$\delta$ER with median elimination method ME-AS [16]. When $k=1$, we denote $k$-$\delta$E and $k$-$\delta$ER, as $\delta$E and $\delta$ER respectively, and compare them with the best arm algorithm ME [5]. We do not experimentally compare to [4] since there is no prior knowledge of $\Delta_k$ in this paper. Note that ME-AS is designed for relative error. To make a fair comparison, given the absolute error bound $\epsilon$, we transform it to $\epsilon/\theta_1$, where $\theta_1$ is the largest mean in the given bandit. $\epsilon/\theta_1$ is used as the equivalent relative error bound in ME-AS. As proved in Lemma 1, without compromising correctness, we can adjust the elimination indicator in $k$-$\delta$E (Line 10 in Algorithm 1). We change $3/4\epsilon$ to $1/2\epsilon$ and set $Q$ to be $\frac{8}{\epsilon^2}$ in our implementation, to gain even better performance.

Without loss of generality, we test our algorithms and competitors on arms following independent Bernoulli distributions with various means. We set the number of total arms to be $n=2000$. We test the methods on three synthetic datasets, as follows:

- Uniform: $\theta_i \sim \text{Unif}[0,1]$. The mean of arms, $\theta_i$, are uniformly distributed in $[0,1]$.
- Normal: $\theta_i \sim TN(0.5,0.2)$. Each $\theta_i$ is generated from a truncated normal distribution with mean 0.5, the standard deviation 0.2 and the support $[0,1]$.
- Segment: $\theta_i = 0.5$ for $i=1,\cdots,k$ and $\theta_i = 0.4$ for $i=k+1,\cdots,n$.

Default parameter values are set as: $\delta=0.1$, and $R=2$. For each setting, the results are averaged over 100 repeated runs. As shown later, ME-AS can be very costly and takes too long time to obtain their average performance over 100 runs, so we terminate them when time is up and report the average obtained. We vary $\epsilon$ from 0.01 to 0.1, while keeping other parameters unchanged.

|        | Algorithm | Uniform | Normal | Segment |
|--------|-----------|---------|--------|---------|
|        | ME        | 11      | 11     | 11      |
| $k = 1$ | $\delta$E | 2.2     | 3.4    | 3.9     |
|        | $\delta$ER | 2      | 2      | 2       |
|        | ME-AS     | 6       | 6      | 6       |
| $k = 20$ | $k$-$\delta$E | 2.1 | 3.0    | 3.8     |
|        | $k$-$\delta$ER | 2  | 2      | 2       |

Table 4: Number of rounds performed.

| Dataset | Algorithm | Rounds | Query Cost |
|---------|-----------|--------|------------|
| Normal  | EG-$\delta$E | 21  | $1.4 \times 10^8$ |
|         | [8]       | 36     | $6.7 \times 10^9$ |
|         | [17]      | $0.9 \times 10^8$ | $0.9 \times 10^8$ |
| Uniform | EG-$\delta$E | 27  | $2.8 \times 10^9$ |
|         | [8]       | 59     | $1.2 \times 10^{11}$ |
|         | [17]      | $2.4 \times 10^9$ | $2.4 \times 10^9$ |
| Segment | EG-$\delta$E | 6   | $5.6 \times 10^7$ |
|         | [8]       | 24     | $1.3 \times 10^{10}$ |
|         | [17]      | $2.2 \times 10^8$ | $2.2 \times 10^8$ |

Table 5: Exact top-$k$ arms: rounds and query cost.

For the best arm selection ($k = 1$), Figure 1 reports the query cost (i.e., total number of pulls) for $\delta$E, $\delta$ER, and ME on the three datasets. Both $\delta$E and $\delta$ER outperform ME significantly for all the $\epsilon$ values on the three datasets. $\delta$E is about 100 times faster than ME, while $\delta$ER is about 10 times faster than ME. $\delta$E is faster than $\delta$ER since $\delta$ER has hard round limit $R = 2$, while $\delta$E does not has such constraint. The first row of Table 4 shows the number of rounds actually used by each method. $\delta$ER strictly uses only two rounds limited by $R$, and $\delta$E needs slightly more rounds, while ME requires 11 rounds that is several times more. For the top-$k$ arm selection ($k = 20$), Figure 2 reports the query cost for $k$-$\delta$E, $k$-$\delta$ER, and ME-AS when varying $\epsilon$. $k$-$\delta$ER is about 100 times faster than ME-AS, and $k$-$\delta$E is about 1000 times faster. The second row of Table 4 shows the number of rounds used per method. Our methods use far fewer rounds. In summary, our $k$-$\delta$E, $k$-$\delta$ER outperform ME and ME-AS with a huge performance gap.

### 4.2 Experimental Results for Exact Top-$k$ Arm Identification

We evaluate our algorithm for exact top-$k$ arm identification. Our algorithm choose [8] as framework, since [14, 13] only focus on theory part and have big constants. We call our algorithm EG-$\delta$E (Exponential-Gap + $\delta$E), and compare it with Elimination based [8] and UCB based [17] algorithms. Default parameter is set as: $\delta = 0.1$. We set [17]'s parameters following their experimental setting. Other experimental settings are same as the PAC-top-$k$ problems.

Table 5 reports the query and round cost for different methods. Compare with [8], EG-$\delta$E uses fewer rounds and is up to 250 times faster than [8] with respect to query cost. Compare with [17], EG-$\delta$E uses significantly fewer rounds while keeps the query cost on same order.

## 5 Related Work

**Instance-independent arm selection.** Top-$k$ arm selection is first studied under the setting of instance-independent. All such existing works [5, 18, 6, 16, 3, 14] are designed for worst case query complexity, and need $\Theta(\log \frac{n}{k})$ rounds, which is inferior to ours. Median Elimination [5] finds the best arm (when $k = 1$) with query complexity $O(\frac{n}{\epsilon^2} \log \frac{1}{\delta})$ under PAC bound, matching the lower bound in [18]. Our top-$k$ algorithms can be easily handle best arm selection by simply setting $k = 1$. We use the same top-$k$ arm definition as [16], which requires that, with high probability, the selected $i^{th}$-top arm has mean greater than $\theta_{i^*} - \epsilon$, for all $i \in [1, k]$, where $\theta_{i^*}$ is $i^{th}$ largest mean in the whole bandit. Explore-$k$ metric is studied in [6]: with high probability, all the $k$ selected arms have mean greater than $\theta_{k^*} - \epsilon$, where $\theta_{k^*}$ is $k^{th}$ largest mean in the whole bandit. Our metric is tighter than

Explore-$k$ metric, and thus our algorithms can also apply to solve Explore-$k$ problem. Another metric was considered in [3], where the identified $k$ arms can have at most $k\epsilon$ regret in total. [14] studies multi-armed bandit problem under matroid constraints. All these works are elimination based.

**Instance-dependent arm selection.** The query complexity of instance-optimal algorithms (e.g., [4, 7, 13, 15, 8, 10, 9]) is closely tied to the bandit instance and is better than the worst case complexity for 'easy' bandit instances. Some of them [7, 13, 15, 8] are elimination-based, and use instance-independent algorithms like [6] and [5] as a sub-procedure to eliminate the arms. Due to the usage of instance-independent algorithms, in the worst case, each iteration of these instance-dependent algorithms needs $\log n$ rounds. Thus the total round complexity is $O(\log n \cdot \log \Delta_k^{-1})$. Another instance-dependent approach is based on upper or lower confidence bounds (UCB or LUCB), e.g., [10] [9]. With respect to query complexity, UCB methods require a $\log n$ factor, while it is $\log k$ in the lower bound. For round complexity, UCB methods need a huge number of rounds since their round complexity is proportional to the query complexity due to their nature of fully adaptiveness.

**Variant settings on limited rounds.** Under the delayed feedback setting, the reward of pulling an arm in round $\tau$ is delayed to be shown in later round $\tau + t$ [19, 20]. Our methods can simulate this setting when taking an appropriately high value of $t$. Most of the existing works focus on regret minimization rather than top-$k$ arms. Some works [11, 21, 22] investigate the batches arm problem. [11] only considers the regret minimization. [21] only allows to pull an arm once per round; the number of rounds required is $\Omega(\log n)$. In [22], within a round, there are limits for both the number of total pulls and the number of pulls per arm. Its rounds in the worst case is at least $\Omega(\log n)$.

# 6   Conclusion

We study the problems of top-$k$ arm selection in adaptive round model, and propose algorithms that achieve the near-optimal query complexity and match the lower bound of round complexity. In practice, our algorithms outperform existing methods in terms of query cost and round complexity.

# Acknowledgement

This research was supported by National Natural Science Foundation of China (No.U1605251) and by the National University of Singapore under SUG grant R-252-000-686-133.

## Footnotes

[2]All logarithms(e.g., $\log_b^*(n)$) in this paper are to base $b$.

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
