[Supplementary Material]

## A    Concentration Inequalities

**Hoeffding's inequality [23]**:

Let $X_1, \cdots, X_n$ be independent with $u_i = \mathbf{E}[X_i]$, $X_i \in [0,1]$, for all $1 \le i \le n$. Then,

$$\Pr\left(\frac{\sum(X_i - u_i)}{n} \ge t\right) \le \exp[-2n \cdot t^2], \tag{3}$$

and

$$\Pr\left(\frac{\sum(X_i - u_i)}{n} \le -t\right) \le \exp[-2n \cdot t^2], \tag{4}$$

**Markov's inequality**: If $X$ is a non-negative random variable and $a > 0$, then the probability that $X$ is at least $a$ is at most the expectation of $X$ divided by $a$:

$$\Pr(X \ge a) \le \frac{\mathbf{E}[X]}{a}. \tag{5}$$

## B    Proofs of Lemmas in Section 2

**Proof of lemma 1**:

*Proof.* First, we prove that at any iteration $r$, $\Pr[\hat{\theta}_{i^*} \ge \theta_{i^*} - \epsilon/8] \ge 1 - \frac{\delta}{2k}$. In the first iteration (i.e., $r = 1$), $S_r$ is exactly the same as the input arm set $S$ (i.e., $S_1 = S$), and Algorithm 1 samples at least $\frac{32}{\epsilon^2} \log \frac{k}{\delta_1}$ times for every arm in $S_1$. Therefore, every arm $i^*$ is initialized for $\hat{\theta}_{i^*}$. Based on Hoeffding bound, we have $\hat{\theta}_{i^*} \ge \theta_i^* - \epsilon/8$ with probability at least $1 - \frac{\delta_1}{k}$, for the case of $r = 1$.

Assume that in iteration $r - 1$, $\hat{\theta}_{i^*} \ge \theta_{i^*} - \epsilon/8$ holds. Then, at iteration $r$, if $\hat{\theta}_{i^*}$ is not updated, $\hat{\theta}_{i^*} \ge \theta_{i^*} - \epsilon/8$ still holds. Otherwise, by Hoeffding's inequality, in iteration $r$,

$$\Pr[\hat{\theta}_{i^*} \ge \theta_{i^*} - \epsilon/8] \ge 1 - \exp[-(32/\epsilon^2)\log(k/\delta_r) \cdot 2(\epsilon/8)^2] \ge 1 - \delta_r/k. \tag{6}$$

By the union bound, for all iterations,

$$\Pr[\hat{\theta}_{i^*} \ge \theta_{i^*} - \epsilon/8] \ge 1 - \sum_{r=1} \delta_r/k \ge 1 - \sum_{r=1} \frac{\delta}{2k \cdot 2^r} \ge 1 - \frac{\delta}{2k}. \tag{7}$$

Applying the union bound again, we have that with probability $1 - \delta/2$, for all $i \in [1, k]$,

$$\hat{\theta}_{i^*} \ge \theta_{i^*} - \epsilon/8. \tag{8}$$

Let $j_r$ be an arm inserted into $S'$ in iteration $r$. Since we have re-sample $Q_r$ times for $j_r$, the estimation in Line 8 of Algorithm 1 is unbiased. By Hoeffding's inequality,

$$\Pr[\hat{\theta}_{j_r} \le \theta_{j_r} + \epsilon/8] \ge 1 - \delta_r/k. \tag{9}$$

Since we insert at most $k$ values to $S'$, by the union bound, we have that for every arm $j_r$ inserted into $S'$,

$$\Pr[\hat{\theta}_{j_r} \le \theta_{j_r} + \epsilon/8] \ge 1 - \delta_r. \tag{10}$$

Applying the union bound again, we have that for each iteration $r$,

$$\Pr[\hat{\theta}_{j_r} \le \theta_{j_r} + \epsilon/8] \ge 1 - \sum_{r=1} \delta_r \ge 1 - \delta/2. \tag{11}$$

Let $i^o$ be the arm with the $i^{th}$ largest value in $S'$. Assume that both Eq. (8) and Eq. (11) hold. By Eq. (8), there are at least $i$ arms whose empirical values are greater than $\theta_i - \epsilon/8$. From Line 10 of Algorithm 1, we have

$$\theta_{i^o} + \epsilon/8 \ge \hat{\theta}_{i^o} \ge \min\{\hat{\theta}_1, \cdots, \hat{\theta}_{i^*}\} - 3/4\epsilon \ge \hat{\theta}_{i^*} - 3/4\epsilon \ge \theta_{i^*} - 3/4\epsilon - 1/8\epsilon. \tag{12}$$

From above equation, we have that with probability $1 - \delta$, for all $i \in [1, k]$, $\theta_{i^o} \ge \theta_{i^*} - \epsilon$.

Similarly, we can prove the second part of Lemma 1, by replacing $\epsilon/8 \leftarrow (\epsilon - \epsilon_1)/2$, $32/(\epsilon^2) \leftarrow 2/(\epsilon - \epsilon_1)^2$ and $3/4\epsilon \leftarrow \epsilon_1$. □

**Proof of Lemma 2**

*Proof.* For $S_r$, let $\theta_1, \cdots, \theta_{|S_r|}$ be the decreasing order of arm's average reward. In Line 6 of Algorithm 1, if $k' = |S_r|$, then $S_{r+1} = \varnothing$ and the lemma easily follows. Hence, we focus on proving the hard case, i.e., when $k' = k$ in Line 6 of Algorithm 1. Let $\lceil (\frac{\delta_r}{k})^{\beta_r} \cdot |S_r| \rceil + k - 1 = m$ and $\theta_m = u$. We divide $[0, \theta_1]$ into three parts: $[0, u - 3\epsilon/8)$, $[u - 3\epsilon/8, u)$ and $[u, \theta_1]$. And denote $A_1, A_2, A_3$ be the sets of arms in $S_r$ whose average means are in $[0, u - 3\epsilon/8)$, $[u - 3\epsilon/8, u)$ and $[u, \theta_1]$, respectively. Let $X_{A_i}$ be the number of arms in $A_i$ whose empirical values updated at iteration $r$ are greater than $u - 3\epsilon/16$, where $i = 1, 2, 3$. Furthermore, let $b_r$ be the arm with the smallest mean that is inserted into $S'$ in iteration $r$. We prove that with high probability, both following conditions hold.

1: $b_r \notin A_1$;

2: $S'(k) + 3\epsilon/4 \geq u + 3\epsilon/16$.

We first prove Condition 1. By Hoeffding's inequality, for $i \in A_1$,

$$\Pr[\hat{\theta}_i \leq \theta_i + 3\epsilon/16] \geq 1 - \exp\left(-2\epsilon^2/(\frac{16}{3})^2 \cdot (57/\varepsilon^2) \cdot \beta_r \log(\frac{k}{\delta_r})\right) \geq 1 - (\frac{\delta_r}{k})^{4\beta_r}. \quad (13)$$

From Eq. (13), for $i \in A_1$, with probability $1 - (\frac{\delta_r}{k})^{4\beta_r}$, $\hat{\theta}_i \leq \theta_i + 3\epsilon/16 \leq u - 3\epsilon/16$. This implies $\mathbf{E}[X_{A_1}] \leq (\frac{\delta_r}{k})^{4\beta_r} \cdot |S_r|$. From Markov's inequality:

$$\Pr[X_{A_1} \geq (\frac{\delta_r}{k})^{2\beta_r}/4|S_r|] \leq \frac{\mathbf{E}[X_{A_1}]}{(\frac{\delta_r}{k})^{2\beta_r}/4|S_r|} \leq \frac{(\frac{\delta_r}{k})^{4\beta_r} \cdot |S_r|}{(\frac{\delta_r}{k})^{2\beta_r}/4|S_r|} \leq 4(\frac{\delta_r}{k})^{2\beta_r}. \quad (14)$$

Let $\xi_1$ be the event $X_{A_1} \leq (\frac{\delta_r}{k})^{2\beta_r}/4|S_r|$, then $\Pr[\xi_1] \geq 1 - 4(\frac{\delta_r}{k})^{2\beta_r}$.

On the other hand, by Hoeffding's inequality, for $i \in A_3$,

$$\Pr[\hat{\theta}_i \geq \theta_i - 3\epsilon/16] \geq 1 - \exp\left(-\epsilon^2 \cdot \frac{9}{256} \cdot (57/\varepsilon^2) \cdot \beta_r \log(1/\frac{\delta_r}{k})\right) \geq 1 - (\frac{\delta_r}{k})^{4\beta_r}. \quad (15)$$

By the above equation, for $i \in A_3$, $\Pr[\hat{\theta}_i \leq u - 3\epsilon/16] \leq (\frac{\delta_r}{k})^{4\beta_r}$. We divide $A_3$ into two parts $B_1$ and $B_2$, where $|B_1| = k - 1$ and $|B_2| = \lceil (\frac{\delta_r}{k})^{\beta_r}|S_r| \rceil$. Then

$$\Pr[X_{B_1} = k - 1] \geq 1 - (\frac{\delta_r}{k})^{4\beta_r} \cdot (k - 1) \geq 1 - (\frac{\delta_r}{k})^{3\beta_r}, \quad (16)$$

and

$$\begin{aligned}
\Pr[X_{B_2} < \lceil (\frac{\delta_r}{k})^{\beta_r}|S_r|/2 \rceil] &\leq \left(\frac{\lceil (\frac{\delta_r}{k})^{\beta_r}|S_r| \rceil}{\lceil (\frac{\delta_r}{k})^{\beta_r}|S_r| \rceil - \lceil (\frac{\delta_r}{k})^{\beta_r}|S_r|/2 \rceil + 1}\right) \cdot ((\frac{\delta_r}{k})^{4\beta_r})^{\lceil (\frac{\delta_r}{k})^{\beta_r}|S_r| \rceil - \lceil (\frac{\delta_r}{k})^{\beta_r}|S_r|/2 \rceil + 1} \\
&\leq 2^{\lceil (\frac{\delta_r}{k})^{\beta_r}|S_r| \rceil} \cdot ((\frac{\delta_r}{k})^{4\beta_r})^{\lceil (\frac{\delta_r}{k})^{\beta_r}|S_r|/2 \rceil} \\
&\leq 2^{\lceil (\frac{\delta_r}{k})^{\beta_r}|S_r| \rceil} \cdot ((\frac{\delta_r}{k})^{2\beta_r})^{\lceil (\frac{\delta_r}{k})^{\beta_r}|S_r|/2 \rceil} \cdot ((\frac{\delta_r}{k})^{2\beta_r})^{\lceil (\frac{\delta_r}{k})^{\beta_r}|S_r|/2 \rceil} \\
&\leq 2^{\lceil (\frac{\delta_r}{k})^{\beta_r}|S_r| \rceil} \cdot (1/16)^{\lceil (\frac{\delta_r}{k})^{\beta_r}|S_r| \rceil} \cdot ((\frac{\delta_r}{k})^{2\beta_r})^{\lceil (\frac{\delta_r}{k})^{\beta_r}|S_r|/2 \rceil} \\
&\leq (\frac{\delta_r}{k})^{2\beta_r}.
\end{aligned}$$
$$(17)$$

Let $\xi_2$ be the event that $X_{A_3} \geq \lceil (\frac{\delta_r}{k})^{\beta_r}|S_r|/2 \rceil + k - 1$, then $\Pr[\xi_2] \geq 1 - (\frac{\delta_r}{k})^{2\beta_r} - (\frac{\delta_r}{k})^{3\beta_r}$.

If both $\xi_1$ and $\xi_2$ hold, then there are more than $\lceil (\frac{\delta_r}{k})^{\beta_r}|S_r|/2 \rceil + k - 1$ arms with empirical value greater than $u - 3\epsilon/16$. By the union bound, $\Pr[\xi_1 \cap \xi_2] \geq 1 - 4(\frac{\delta_r}{k})^{2\beta_r} - (\frac{\delta_r}{k})^{2\beta_r} - (\frac{\delta_r}{k})^{3\beta_r} = 1 - 5(\frac{\delta_r}{k})^{2\beta_r} - (\frac{\delta_r}{k})^{3\beta_r}$.

Assume that both $\xi_1$ and $\xi_2$ hold, then in iteration $r$

$$
\begin{aligned}
\Pr(b_r \in A_1) &\leq \frac{(\frac{\delta_r}{k})^{2\beta_r}/4|S_r|}{\lceil(\frac{\delta_r}{k})^{\beta_r}|S_r|/2\rceil + k - 1} + (1 - \frac{(\frac{\delta_r}{k})^{2\beta_r}/4|S_r|}{\lceil(\frac{\delta_r}{k})^{\beta_r}|S_r|/2\rceil + k - 1})\frac{(\frac{\delta_r}{k})^{2\beta_r}/4|S_r|}{\lceil(\frac{\delta_r}{k})^{\beta_r}|S_r|/2\rceil + k - 2} + \cdots \\
&\leq \frac{(\frac{\delta_r}{k})^{2\beta_r}/4|S_r|}{\lceil(\frac{\delta_r}{k})^{\beta_r}|S_r|/2\rceil + k - 1} + \frac{(\frac{\delta_r}{k})^{2\beta_r}/4|S_r|}{\lceil(\frac{\delta_r}{k})^{\beta_r}|S_r|/2\rceil + k - 2} + \cdots \\
&\leq \sum_{i=1}^{k} \frac{(\frac{\delta_r}{k})^{2\beta_r}/4|S_r|}{\lceil(\frac{\delta_r}{k})^{\beta_r}|S_r|/2\rceil + i - 1} \\
&\leq k \cdot (\frac{\delta_r}{k})^{\beta_r}/2.
\end{aligned}
$$
(18)

Therefore, Condition 1 holds.

Next, we assume that Condition 1 holds and focus on proving Condition 2. In Algorithm 1, since we re-sample $Q_r$ times for each arm $j_r \in S_{k'}^r$,

$$
\Pr(\hat{\theta}_{j_r} \geq \theta_{j_r} - 3\epsilon/16) \geq 1 - (\frac{\delta_r}{k})^{4\beta_r}.
$$
(19)

Let $\hat{\theta}_{min}$ be the smallest value inserted into $S'$ in iteration $r$. By the union bound and Eq. (19), with probability $1 - k \cdot (\frac{\delta_r}{k})^{4\beta_r}$,

$$
\hat{\theta}_{min} \geq \theta_{b_r} - 3\epsilon/16 \geq u - 9\epsilon/16.
$$
(20)

If $\hat{\theta}_{min} \geq u - 9\epsilon/16$ holds, then $S'(k) + 3\epsilon/4 \geq \hat{\theta}_{min} + 3\epsilon/4 \geq u + 3\epsilon/16$. Therefore, Condition 2 holds.

Similar to Eq. (13) and (14), we can prove that with probability $1 - 4(\frac{\delta_r}{k})^{2\beta_r}$, there are less than $(\frac{\delta_r}{k})^{2\beta_r}/4|S_r|$ arms in $A_1$ and $A_2$ whose empirical values are larger than $u + 3\epsilon/16$. This means that under Conditions 1 and 2, with probability $1 - 4(\frac{\delta_r}{k})^{2\beta_r}$, $|S_{r+1}| \leq (\frac{\delta_r}{k})^{2\beta_r}/4|S_r| + \lceil(\frac{\delta_r}{k})^{\beta_r}|S_r|\rceil + k - 1 - k$. Note that the $-k$ term is due to the factor that $S_{k'}^r$ is selected from top-$\lceil(\frac{\delta_r}{k})^{\beta_r}|S_r|/2\rceil + k' - 1\rceil$ arms but is eliminated in Line 10 of Algorithm 1.

Applying the union bound, we have

$$
\Pr[|S_{r+1}| \leq (\frac{\delta_r}{k})^{2\beta_r}/4|S_r| + \lceil(\frac{\delta_r}{k})^{\beta_r}|S_r|\rceil - 1] \geq 1 - 5(\frac{\delta_r}{k})^{2\beta_r} - (\frac{\delta_r}{k})^{3\beta_r} - k(\frac{\delta_r}{k})^{\beta_r}/2 - k(\frac{\delta_r}{k})^{4\beta_r} - 4(\frac{\delta_r}{k})^{2\beta_r}
$$
(21)

Note that $\beta_r \geq 1$, $\delta_r \leq \delta/4 \leq 1/16$, we have

$$
\Pr[|S_{r+1}| \leq \lceil 2(\frac{\delta_r}{k})^{\beta_r}|S_r|\rceil - 1] \geq 1 - \frac{2\delta_r}{k} \geq 1 - 2\delta_r.
$$
(22)

$\square$

**Proof of Lemma 3**

*Proof.* Proof of Part 1: Assume that Algorithm 1 terminates at iteration $r'$. Note that in Algorithm 1, each iteration has two rounds, and hence, $R' = 2r'$. We prove $r' \leq \log_{\frac{k}{\delta}}^*(n)$.

For $r \leq r'$, by Lemma 2, with probability $1 - 2\delta_r$, $|S_{r+1}| \leq \lceil 2 \cdot (\delta_r/k)^{\beta_r}|S_r|\rceil - 1 \leq 2 \cdot (\delta_r/k)^{\beta_r}|S_r|$. This implies that with probability $1 - 2\delta_r$,

$$
\beta_{r+1} \geq \frac{|S_r|}{2|S_{r+1}|} \cdot \beta_r \geq \frac{|S_r|}{4 \cdot (\delta_r/k)^{\beta_r}|S_r|} \cdot \beta_r = \frac{\beta_r}{4 \cdot (\delta_r/k)^{\beta_r}} \geq \frac{1}{(\delta/k)^{\beta_r}}.
$$
(23)

Thus, with probability $1 - 2(\delta_1 + \cdots + \delta_r)$,

$$
\beta_{r+1} \geq (\frac{k}{\delta})^{\beta_r} \geq (\frac{k}{\delta})^{(\frac{k}{\delta})^{\beta_{r-1}}} \geq \cdots \geq \underbrace{\frac{k}{\delta}^{\frac{k}{\delta}^{\cdot^{\cdot^{\frac{k}{\delta}}}}}}_{r}
$$
(24)

For integer $m \geq \log^*_{\frac{k}{\delta}}(n)$, we have $n \leq \underbrace{\frac{k}{\delta}^{\frac{k}{\delta} \cdot^{\cdot^{\cdot \frac{k}{\delta}}}}}_{m}$. If $m \leq r'$, by Eq. (24), we have that with probability $1 - 2(\delta_1 + \cdots + \delta_{m-1})$

$$\beta_m \geq \underbrace{\frac{k^{\frac{k}{\delta} \cdot^{\cdot^{\cdot \frac{k}{\delta}}}}}{\delta}}_{m-1}. \tag{25}$$

Assume that Eq. (25) holds. Then, by Lemma 2, with probability $1 - 2\delta_m$, $|S_{m+1}| \leq 2 \cdot (\delta_m/k)^{\beta_m}|S_m| \leq |S_m|/(2 \cdot \underbrace{\frac{k}{\delta}^{\frac{k}{\delta} \cdot^{\cdot^{\cdot \frac{k}{\delta}}}}}_{m}) \leq |S_m|/(2n) < 1$. This implies $S_{r+1} = \varnothing$. Applying the union bound, we have that w.p.

$$1 - \sum_{i=1} 2\delta_i \geq 1 - \delta, \tag{26}$$

$R' = 2r' \leq 2\log^*_{\frac{k}{\delta}}(n)$.

Proof of Part 2: Let $\rho = \log^*_{\frac{k}{\delta}}(n)$. Given positive integer $l$, we prove w.p. $1 - \frac{\delta}{2^{l-1}}$, $r' \leq l\rho$. Similar to Eq. (23) and (24), we have that with probability $1 - 2(\delta_{(l-1)\rho+1} + \delta_{(l-1)\rho+2} + \cdots + \delta_{l\rho-1})$,

$$\beta_{l\rho} \geq \frac{1}{(\delta/k)^{\beta_{l\rho-1}}} \geq \cdots \geq \frac{1}{(\delta/k)^{\beta_{(l-1)\rho+1}}}^{\frac{k}{\delta} \cdot^{\cdot^{\cdot \frac{k}{\delta}}}} \geq \underbrace{\frac{k^{\frac{k}{\delta} \cdot^{\cdot^{\cdot \frac{k}{\delta}}}}}{\delta}}_{\rho-1} \tag{27}$$

If above equation holds, then with probability $1 - \delta_{l\rho}$, $S_{l\rho+1} \leq 2 \cdot (\delta_{l\rho}/k)^{\beta_{l\rho}}|S_{l\rho}| \leq |S_{l\rho}|/(2 \cdot \underbrace{\frac{k}{\delta}^{\frac{k}{\delta} \cdot^{\cdot^{\cdot \frac{k}{\delta}}}}}_{\rho}) \leq |S_{l\rho}|/(2n) < 1$, which means that $S_{l\rho} = \varnothing$. By the union bound, with probability $1 - 2(\delta_{(l-1)\rho+1} + \delta_{(l-1)\rho+2} + \cdots + \delta_{l\rho}) \geq 1 - \frac{\delta}{2^{(l-1)\rho}}$, $r' \leq l\rho$.

Now, we have

$$\begin{aligned} \mathbf{E}[R'] &\leq 2(1 - \delta) \cdot \rho + 2(\delta - \frac{\delta}{2^\rho}) \cdot (2\rho) + 2(\frac{\delta}{2^\rho} - \frac{\delta}{2^{2\rho}}) \cdot (3\rho) + \cdots \\ &\leq 2\rho + \sum_{i=1}[\frac{2\delta\rho}{2^{(i-1)\rho}}] \\ &\leq 2(1 + 2\delta)\rho = 2(1 + 2\delta)\log^*_{\frac{k}{\delta}}(n) \end{aligned} \tag{28}$$

$\square$

**Proof of Lemma 4**

*Proof.* Proof of Part 1: Let $\xi$ be the event that $\beta_{r+1} = \beta_r \cdot \frac{|S_r|}{2|S_{r+1}|}$ holds for all $r \geq 1$. By Lemma 2, with probability $1 - 2\delta_r$, $|S_{r+1}| \leq [2 \cdot (\delta_r/k)^{\beta_r}|S_r|] - 1 \leq 2 \cdot (\delta_r/k)^{\beta_r}|S_r| \leq \frac{2\delta}{k}|S_r|$. Hence w.p. $1 - 2\delta_r$, $\beta_{r+1} = \beta_r \cdot \frac{|S_r|}{2|S_{r+1}|}$. Applying the union bound, the event $\xi$ holds with probability $1 - \delta$.

Assume that $\xi$ holds. In iteration $r$, the total number of arms pulled by Lines 5 and 7 of Algorithm 1 is $(|S_r| + 1) \cdot \beta_r \cdot Q \cdot \log \frac{k}{\delta_r}$. Note that $\delta \in (0, 1/4)$ and $\beta_{r+1} = \beta_r \frac{|S_r|}{2|S_{r+1}|}$, we have

$$\begin{aligned} N &= \sum_{i=1}[(|S_i| + 1) \cdot \beta_i \cdot Q \cdot \log \frac{k}{\delta_i}] \leq 2\sum_{i=1}[|S_i| \cdot \beta_i \cdot Q \cdot \log \frac{k}{\delta_i}] \\ &\leq 2\sum_{i=1}[\frac{n}{2^{i-1}} \cdot Q \cdot (\log \frac{4k}{\delta} + i)] \\ &\leq 6n \cdot Q + 4nQ \cdot \log \frac{4k}{\delta} \end{aligned} \tag{29}$$

Thus with probability $1 - \delta$, $N \leq 7n \cdot Q \cdot \log \frac{4k}{\delta}$.

Proof of Part 2: Let $N_i$ be the number of samples in iteration $r$. By Lemma 2, with probability $1 - 2\delta_r$, $|S_{r+1}| \leq \lceil 2 \cdot (\delta_r/k)^{\beta_r} |S_r| \rceil - 1 \leq 2 \cdot (\delta_r/k)^{\beta_r} |S_r| \leq \frac{2\delta}{k} |S_r|$. Then, by Algorithm 1, with probability $1 - 2\delta_r$, $\beta_{r+1} \cdot |S_{r+1}| = \beta_r \cdot |S_r|/2$. If $|S_{r+1}| \geq 16$, given that $\delta_r \leq \frac{1}{16}$, we have

$$
\begin{aligned}
\mathbf{E}[N_{r+1}] &= (|S_{r+1}| + 1) \cdot \beta_{r+1} \cdot Q \cdot \log \frac{k}{\delta_{r+1}} \leq \frac{17}{16} |S_{r+1}| \cdot Q \cdot \beta_{r+1} \cdot \log \frac{k}{\delta_{r+1}} \\
&\leq \frac{17}{32} (1 - 2\delta_r) \cdot |S_r| \cdot Q \cdot \beta_r \cdot \log \frac{k}{\delta_{r+1}} + \frac{17}{16} \cdot 2\delta_r \cdot |S_r| \cdot Q \cdot \beta_r \cdot \log \frac{k}{\delta_{r+1}} \\
&\leq (1 - 2\delta_r) \cdot \frac{17 N_r}{32} \cdot \frac{\log(k/\delta_{r+1})}{\log(k/\delta_r)} + \frac{17}{8} \delta_r \cdot N_r \cdot \frac{\log(k/\delta_{r+1})}{\log(k/\delta_r)} \\
&\leq \frac{3}{4} N_r.
\end{aligned}
\tag{30}
$$

Then

$$
\mathbf{E}[N_1 + \cdots + N_{r+1}] \leq \mathbf{E}[N_1] + 3/4\mathbf{E}[N_1] + \cdots + (3/4)^r \mathbf{E}[N_1] \leq 4\mathbf{E}[N_1].
\tag{31}
$$

If $|S_{r+1}| \leq 16$, then

$$
N_{r+1} = (|S_{r+1}| + 1) \cdot \beta_{r+1} \cdot Q \cdot \log \frac{k}{\delta_{r+1}} \leq 2|S_{r+1}| \cdot \beta_{r+1} \cdot Q \cdot \log \frac{k}{\delta_{r+1}} \leq \frac{5}{2} N_r.
\tag{32}
$$

By Lemma 2, with probability $1 - 2\delta_{r+1}$, $|S_{r+2}| \leq \frac{2\delta_{r+1}}{k} |S_{r+1}| - 1 < 1$. Thus

$$
\mathbf{E}[N_{r+2}] \leq 2\delta_{r+1} \cdot \frac{5}{2} N_{r+1} \leq \frac{5}{32} N_{r+1}.
\tag{33}
$$

Hence,

$$
\mathbf{E}[N_{r+1} + N_{r+2} \cdots] \leq \mathbf{E}[N_{r+1}] + \frac{5}{32} \mathbf{E}[N_{r+1}] + \cdots \leq \frac{32}{27} \mathbf{E}[N_{r+1}] \leq \frac{80}{27} N_r \leq \frac{80}{27} N_1.
\tag{34}
$$

Therefore, $\mathbf{E}[N] = \sum_{i=1} \mathbf{E}[N_i] \leq 4N_1 + \frac{80}{27} N_1 \leq 7(n + 1) \cdot Q \cdot \log \frac{k}{\delta_1}$. $\qquad\square$

**Proof of Lemma 5**

*Proof.* We first focus on iteration $r \leq R - 1$. For convenience, we follow the notation used in the proof of Lemma 2. By Eq.(21) in the proof of Lemma 2 , we have that with probability at least $1 - 5(\frac{\delta_r}{k})^{2\beta_r} - (\frac{\delta_r}{k})^{3\beta_r} - k(\frac{\delta_r}{k})^{\beta_r}/2 - k(\frac{\delta_r}{k})^{4\beta_r} - 4(\frac{\delta_r}{k})^{2\beta_r} \geq 1 - \frac{3}{2}\delta_r$, both of the following conditions hold.

1: $b_r \notin A_1$;

2: All the arms' empirical values in $S_r \backslash S_{r+1}$ are smaller than $u + 3/16\epsilon$.

By Hoeffding's inequality, for arm $j$ in iteration $r$, we have

$$
\Pr(\hat{\theta}_j \geq \theta_j - 3\epsilon/16) \geq 1 - (\frac{\delta_r}{k})^{4\beta_r}.
\tag{35}
$$

For $r = 1$, then applying the union bound, for arm set $\{1^*, 2^* \cdots, i^*\}$, with probability $1 - k \cdot (\frac{\delta_1}{k})^{4\beta_1}$, all arm's empirical value greater than $\theta_{i^*} - 3\epsilon/16$. Applying the union bound for all iterations, we have that $w.p.$ at least $1 - \frac{\delta^3}{32}$, there are at least $i$ arms in set $\{1^*, 2^*, \cdots, i^*\}$ whose empirical values are greater than $\theta_{i^*} - 3\epsilon/16$. Let $\xi$ be the above event. Then $\Pr[\xi] \geq 1 - \frac{\delta^3}{32}$.

Without loss of generality, we assume that $i^*$ is the first eliminated arm in $\{1^*, 2^* \cdots, i^*\}$, and that it is eliminated in iteration $r$. Let $j_r$ be an arm inserted into $S'$ in iteration $r$. Assume $\xi$ holds. Combining two conditions together, we have that with probability $1 - \frac{3}{2}\delta_r$,

$$
\theta_{j_r} \geq \theta_{b_r} \geq u - 3/8\epsilon = u + 3/16\epsilon - 9/16\epsilon \geq \hat{\theta}_{i^*} - 9/16\epsilon \geq \theta_{i^*} - 3/4\epsilon.
\tag{36}
$$

In addition, Without loss of generality, we assume that $i^*$ has been eliminated before iteration $R$ and all $x^*$ $(x < i)$ are kept in $S_R(S_R \neq \varnothing)$. In iteration $R$ of Algorithm 3,

$$\beta_R \geq (\frac{k}{\delta})^{\beta_{R-1}} \geq (\frac{k}{\delta})^{(\frac{k}{\delta})^{\beta_{R-2}}} \geq \cdots \geq \underbrace{\frac{k}{\delta}^{\cdot^{\cdot^{\cdot\beta_1}}}}_{R-1} \tag{37}$$

By the definition of $\beta_1$, we have $\lceil (\delta_R/k)^{\beta_R} |S_R|/2 \rceil < 1$. It follows that $(\frac{k}{\delta_R})^{\beta_R} \geq |S_R|/2$. Thus, we have pulled each arm in $S_R$ at least

$$Q \cdot \log(\frac{k}{\delta_R})^{\beta_R} \geq \max\{Q \cdot \log\frac{S_R}{2}, Q \cdot \log\frac{k}{\delta_R}\} \geq \frac{Q \cdot \log\frac{S_R}{2} + Q \cdot \log\frac{k}{\delta_R}}{2} \geq \frac{Q}{2} \log\frac{k \cdot S_R}{2\delta_R} \tag{38}$$

times. Applying Hoeffding's inequality and the union bound, for every $j_R \in S_R$, we have that with probability $1 - 2\delta_R$,

$$\theta_{j_R} - 1/7\epsilon \leq \hat{\theta}_{j_R} \leq \theta_{j_R} + 1/7\epsilon. \tag{39}$$

In Algorithm 3, we pull each arm (in $S'$) $Q \log\frac{4|S'|}{\delta}$ times. By Hoeffding's inequality and the union bound, we have that with probability $1 - \frac{\delta}{4}$, for all $s \in S'$,

$$\theta_s - 1/10\epsilon \leq \hat{\theta}_s \leq \theta_s + 1/10\epsilon. \tag{40}$$

By Eq. (36) and union bound, with probability $1 - \frac{3}{2}\sum_{r=1}\delta_r - \Pr[\xi] \geq 1 - \delta/2$, there are at least $k$ arms in $S'$ whose means are greater than $\theta_{i^*} - 3/4\epsilon$. Combining Eq. (40) and the union bound, we have that for any arm $s$ whose empirical value is top-$i$ in Line 4 of Algorithm 3, with probability $1 - \frac{3\delta}{4}$,

$$\theta_s \geq \hat{\theta}_s - 1/10\epsilon \geq \hat{\theta}_{i^*} - 3/4\epsilon - 1/10\epsilon > \theta_{i^*} - \epsilon. \tag{41}$$

Assume that Eq. (41),(40) and (39) hold. Again, we assume $i^*$ has been eliminated before iteration $R$ and all $x^*(x < i)$ are kept in $S_R(S_R \neq \varnothing)$. Let $x^o$ be the top-$x$ returned arm. If $x^o \in S'$, we consider two cases as follows:

Case 1:$x \geq i$. By Eq. (41), $\theta_{x^o} \geq \theta_{i^*} - \epsilon$.

Case 2:$x < i$. By Eq. (40) and (39), we have

$$\theta_{x^o} + 1/10\epsilon \geq \hat{\theta}_{x^o} \geq \hat{\theta}_{x^*} \geq \theta_{x^*} - 1/7\epsilon. \tag{42}$$

Thus $\theta_{x^o} \geq \theta_{x^*} - \epsilon$.

On the other hand, if the returned arm $x^o \notin S'$, then we have $x^o \in S_R$. We differentiate two cases as follows.

Case 1: $x < i$. Note that $x^* \in S_R$. By Eq. (39), we have

$$\theta_{x^o} \geq \hat{\theta}_{x^o} - 1/7\epsilon \geq \hat{\theta}_{x^*} - 1/7\epsilon \geq \theta_{x^*} - 2/7\epsilon. \tag{43}$$

Case 2: $x \geq i$. We have

$$\theta_{x^o} \geq \hat{\theta}_{x^o} - 1/7\epsilon \geq \theta_{i^*} - 3/4\epsilon - 1/7\epsilon - 1/10\epsilon \geq \theta_{i^*} - \epsilon. \tag{44}$$

Applying the union bound, Eq. (41), (40), and (39) hold with probability at least $1 - \delta$, and hence, the lemma is proved. □

## Proof of Lemma 6

*Proof.* Let $N_i$ be the number of arms pulled in iteration $r \leq R - 1$. For $r \leq R - 2$, we have

$$N_{r+1} = |S_{r+1}| \cdot \beta_{r+1} \cdot Q \cdot \log\frac{k}{\delta_{r+1}} \leq \frac{1}{2}|S_r| \cdot \beta_r \cdot Q \cdot \log\frac{k}{\delta_{r+1}} \leq \frac{3}{4}N_r. \tag{45}$$

Meanwhile, the $R^{th}$ round consumes in two places: Line 2 and Line 4 in Algorithm 3. Let $X$ be the sample cost of Line 2. Similar to Equation 45, we have $X \leq 3/4N_{R-1}$. For Line 4, since $|S'| \leq \log^*_{\frac{k}{\delta}}(n) \cdot k$, it can be proved that the cost of Line 4 is bounded by $O(\frac{n}{\epsilon^2}(\log\frac{k}{\delta} + \text{ilog}^{(R)}_{\frac{k}{\delta}}(n)))$.

Therefore, $N \leq O(\sum_{r=1}^{R-1}N_r) + O(\frac{n}{\epsilon^2}(\log\frac{k}{\delta} + \text{ilog}^{(R)}_{\frac{k}{\delta}}(n))) = O(\frac{n}{\epsilon^2}(\text{ilog}^{(R)}_{\frac{k}{\delta}}(n) + \log\frac{k}{\delta})).$ □

**Elimination Arms**

Work [13] can eliminate most arms below a given threshold by using the Elimination procedure defined in [24] as a subroutine. To bound the round complexity of the elimination procedure in our Theorem 3, we need the Elimination procedure in [24], Page 7, $\delta_r \leftarrow \dfrac{\delta}{10 \cdot \underbrace{\frac{1}{\delta}^{\frac{1}{\delta} \cdot^{\cdot^{\cdot}} \frac{1}{\delta}}}_{r}}$. Then, Lemma 2.4

in [24] still holds and each elimination costs $O(\log^*_{\frac{1}{\delta}}(n))$ rounds.

**Proof of Lemma 8**

*Proof.* We consider the Exponential-Gap-Eliminating process and divide it into two part. The first part is $\epsilon_r \in [1, \Delta_k]$. In this part, since $\epsilon_{r+1} = \epsilon_r/8$, we need at most $\log \Delta_k^{-1}$ iterations. For each iteration, we need to run algorithm 1 and use $\Omega(\log^*_{\frac{1}{\delta}}(n))$ rounds. Thus, for the first part, we need $O(\log \Delta_k^{-1} \cdot \log^*_{\frac{1}{\delta}}(n))$ rounds.

The second part is $\epsilon_r < \Delta_k$. Once $\epsilon_r < \Delta_k/8$, from [13]'s Observation 4.2, we can get with high probability the algorithm will return. Thus, the round complexity of second part is $O(\log^*_{\frac{1}{\delta}}(n))$. $\quad\square$