[Reviews · NeurIPS 2019]

Reviewer 1



------------------------------------------ After author feedback: Thanks to the authors for their response. This is a paper with strong theoretical results which are well supported by the experiments. ------------------------------------------- Original review: Summary: The object of study is the round complexity of (epsilon, delta)-PAC best-k-arms identification. In that particular instance of the best-k-arms bandit problem, an algorithm successively samples arms, with the goal of finally returning a set of k arms, which with probability bigger than 1-delta verifies that the mean or the j^th arm of the set is within epsilon of the j^th highest mean. The focus is on algorithms which have a small number of rounds at which they make decisions: at each such round, the algorithm queries a number of samples, and no adaptive decision is made between rounds. An algorithm is good if the total number of samples it requires is small (sample complexity) and if the number of rounds is small (round complexity). New algorithms and complexity bounds are provided for two settings: either the algorithm is free to choose the number of rounds, or that number is set in advance. An adaptation to exact top-k arm identification (epsilon=0) is also presented. Strengths: - Strong theoretical guarantees: the algorithms have both close to optimal sample complexities and round complexities. These algorithms are the first ones with such round complexities which do not require additional information on the problem. - Convincing experimental evaluation. The new algorithms improve over existing ones by order of magnitude with respect to the round complexity, while having sample complexity of comparable order to the best ones. The results on the exact top-k problem are particularly striking. - Clear presentation of the algorithms, in particular on page 4. Weaknesses: - the description and comparison of the different existing top-k settings could be clearer. For example, some papers require the means of all the k arms to be above the k^th highest mean, while this paper compares each of the k arms to different reference arms. This is mentioned in the related work section at the end of the paper, but the clarity would be enhanced if it was explained in the introduction. - The references to existing results could be more precise. For example the lower bound mentioned line 168 could be cited more precisely. Instead of referring to papers [16] and [14] without more precisions, the authors could write for example [16, Theorem X]. Lines 100-102, a notion of "near-optimality" coming from a previous paper is also referred to in a vague way. Remarks: - I think that the notation log^*(a) is defined, but not log^*_b(a). It took me a moment to realize that this was the logarithm in base b. - Instead of calling the problems 1, 2 and 3, it would be easier for the reader to have them be called by names. Writing "exact-top-k" is not much longer than "problem 3" and is much more descriptive.

Reviewer 2



- The existing algorithms need \log^\star(n) rounds. Given the slow growth of $\log^\star(n)$, trying to further reduce this number does not seem that interesting in practice. This number is smaller than 5 for $n \le 2^{65536}$. - While there might exist differences in the setting, the authors of [4] claim that minimal round complexity scales at least as $\Omega( \log^\star(n))$, so that the authors should clarify how their setting differs from [4], as otherwise their results would simply prove that [4] contains a mistake. - As a minor remark, it could be good for clarity to explicity define the $\log^\star(n)$ function for an arbitrary base (other than 2). - Also, the authors mention that the algorithm proposed in [4] requires the knowledge of a number $\Delta$, such that $\Delta_k > \Delta$. Therefore, a straightforward approch to extend it to the present setting (where $\Delta$ is unknown) would be to apply the algorithm of [4] for \Delta = 1, then check that $\Delta_k > \Delta$ by sampling. It is noted that one can always estimate $\Delta_k$ by sampling all arms enough time (and this takes one "round") If the test fails, do the same for $\Delta = \gamma$, $\Delta = \gamma^2$ (where $\gamma < 1$) and so on and so forth until the test succeeds. Would such an approach fail and if not why ?

Reviewer 3



This paper focused mainly on the PAC top-k-subset selection problem in an adaptive round model, which means the authors are not only interested in the sample complexity of the proposed algorithm, but also try to achieve minimal number of rounds used. The motivation of this setting has been stated and the main text is globally well written. The algorithms are elimination-based that try to kill as much arms as they can in each round to reduce the round complexity. It appears that the query complexity in the fixed-confidence setting stays optimal and shows a good empirical performance as well. I have a major technique concern, however, regarding the proof of Lemma 1. In equation 7, you are using a union bound over all iterations r. But the Hoeffding bound only holds for iteration r if i^* has been updated at this iteration. What if it is not updated at this iteration? \hat{\theta_i^*} >= \theta_i^* - \epsilon/8 holds for iteration r only when it holds for iteration r-1, but that cannot be true from the very beginning. Maybe I am missing something, but I really need more clarification on this part, which seems to be a crucial lemma in this paper. Otherwise, regarding the experiments, have you ever thought of comparing \deltaE to some other fixed confidence BAI algorithms and \deltaER to some fixed budget BAI algorithms. Of course, it is not comparable straightforwardly, but have you thought of proposing some naive adaptation of those algorithms to your setting? Minor: - I would suggest the authors to make the motivation more clear as you only stated the use of muliple testing for several applications but didn't really explain why we need to do so in the real world - The Exponential-Gap-Elimination alogorithm could have been detailed in the paper (or appendix), otherwise the paper is not self-contained Update after author feedback: I am convinced by the author's feedback regarding the proof, and thus increased the score. In summary, this paper has some strong theoretical results that worth being published in NeurIPS.

[Author Response · NeurIPS 2019]

We thank the reviewers for the insightful and constructive comments. In what follows, we provide our response to the
major concerns raised.

**Reviewer 1:** We agree with the presentation and reference issues raised, and will revise the paper as advised.

**Reviewer 2, Comments 1 & 2:** We agree that $\log^*(n)$ is a small number and further reducing it is not interesting.
However, we note that our focus is not to reduce the round number below $\log^*(n)$, but to achieve $O(\log^*(n))$ rounds
*without relying on the unrealistic assumptions made in [4]*. In particular, [4] assumes that $\Delta_k$ (i.e., the difference
between the means of the $k^{th}$ and $(k+1)^{th}$ largest arms) is known, which is seldom the case in practice given that
the means of all arms are unknown in advance. In contrast, our algorithms do not require any prior knowledge of $\Delta_k$;
we allow users to choose an error parameter $\epsilon \in (0, 1)$ to strike a trade-off between accuracy and efficiency. In our
submission, we discuss the above issues in Lines 80-88 and 103-106. We also note that our $O(\log^*_{\frac{k}{\delta}}(n))$ result does not
contradict the $\Omega(\log^*(n))$ lower bound in [4], since the latter regard $k$ and $\delta$ as constants, whereas we consider them to
be variables. If we also consider $k$ and $\delta$ to be constants, then our result would match the lower bound in [4].

**Reviewer 2, Comment 3:** We will explicitly define $\log^*_b(n)$ as advised.

**Reviewer 2, Comment 4:** We have considered the suggested approach (which tests $\Delta = 1, \gamma, \gamma^2, \cdots$), but found
that its complexity is inferior to ours, as explained in the following. Assume that the approach stops testing when
$\Delta = \gamma^{2t}$. For the PAC setting (see Problem 1 in our submission), since $\gamma^{2t} < \Delta_k$, the suggest approach has to call
the algorithm in [4] $O(\log \Delta_k^{-1})$ times, each of which requires $\log^*(n)$ rounds. Therefore, its round complexity is
$O(\log \Delta_k^{-1} \cdot \log^* n)$, which is inferior to our $O(\log^*_{\frac{k}{\delta}}(n))$ result. For the exact top-$k$ setting (see Problem 3 in our
submission), let us consider a bandit instance where we have (i) $k$ arms with means $\theta$, (ii) $n - k$ arms with means
$\theta - \Delta_k$, and $\gamma^t = 2\Delta_k$ (i.e., $\gamma^{2t} = 4\Delta_k^2$). If the suggested approach stops testing at $\Delta = \gamma^{2t}$, its query complexity is
at least $O\left(\frac{n}{\Delta_k^4} \log \frac{k}{\delta}\right)$, which is inferior to our query complexity $O\left(\frac{n}{\Delta_k^2} \log \frac{k \cdot \log \Delta_k^{-1}}{\delta}\right)$.

**Reviewer 3, Comment 1:** For the proof of Lemma 1, we note that $\hat{\theta}_{i^*} \geq \theta_i^* - \epsilon/8$ holds with high probability even at
the very beginning. In particular, in the first iteration (i.e., $r = 1$), $S_r$ is exactly the same as the input arm set $S$ (i.e.,
$S_1 = S$), and Algorithm 1 samples at least $\frac{32}{\epsilon^2} \log \frac{k}{\delta_1}$ times for every arm in $S_1$. Therefore, every arm $i^*$ is initialized
for $\hat{\theta}_{i^*}$. Based on Hoeffding bound, we have $\hat{\theta}_{i^*} \geq \theta_i^* - \epsilon/8$ with probability at least $1 - \frac{\delta_1}{k}$, for the case of $r = 1$.

The case for $r > 1$ follow from an induction on $r$, as shown in Lines 362-377 in our supplementary material.
Specifically, suppose that $\hat{\theta}_{i^*} \geq \theta_i^* - \epsilon/8$ holds in the $(r-1)$-th iteration. If $\hat{\theta}_{i^*}$ is NOT updated in the $r$-th iteration,
then $\hat{\theta}_{i^*} \geq \theta_i^* - \epsilon/8$ remains. Meanwhile, if $\hat{\theta}_{i^*}$ is updated in the $r$-th iteration, then we can apply the Hoeffding bound
to show that after the update, $\hat{\theta}_{i^*} \geq \theta_i^* - \epsilon/8$ holds with at least $1 - \frac{\delta_r}{k}$ probability. By the union bound, the probability
that $\hat{\theta}_{i^*} \geq \theta_i^* - \epsilon/8$ holds in all iterations is at least $1 - \frac{\delta}{2k}$ (see Eq. (7) in our supplementary material).

We will revise the proof of Lemma 1 to avoid confusions over the cases of (i) $r = 1$ and (ii) $r > 1$ and $\hat{\theta}_{i^*}$ is not updated
in the $r$-th iteration.

**Reviewer 3, Comment 2:** Regarding the comparison between $\delta$E and other fixed confidence BAI algorithms: we have
actually done such a comparison in our submission (see Figures 1 and 2). In particular, we compare $\delta$E and $k$-$\delta$E
against ME [5] and ME-AS [6], both of which are state-of-the-art methods for fixed confidence instance-independent
BAI. Our results demonstrate that $\delta$E (resp. $k$-$\delta$E) significantly outperforms ME (resp. ME-AS) in terms of query cost.
In addition, in Section 4.2, we use $\delta$E as a subroutine to construct an algorithm (referred to as EG-$\delta$E) for exact (instead
of PAC) top-$k$ arm identification, and we show that it outperforms the state-of-the-art elimination-based method [8] and
UCB-based method [17].

Meanwhile, for $\delta$ER, we find it difficult to compare it with fixed budget BAI algorithms, due to the significant difference
in the number of rounds that they require. Specifically, existing fixed budget BAI algorithms (e.g., [8]) require at least
$\log(n)$ rounds, whereas $\delta$ER requires *at most* $\log^*(n)$ rounds. This makes it difficult to identify a setting of round
numbers to conduct a fair comparison of the algorithms.

**Reviewer 3, Comments 3 & 4:** We will clarify the motivation for multiple testing and detail the Exponential-Gap-
Elimination algorithm as advised.

[Meta-Review · NeurIPS 2019]

The reviewers all agree on the interest and quality of this work, and I agree. They have a few suggestions which the authors might want to follow so as to provide the best possible final version.